# Phytochemical Profile of *Trigonella caerulea* (Blue Fenugreek) Herb and Quantification of Aroma-Determining Constituents

**DOI:** 10.3390/plants12051154

**Published:** 2023-03-03

**Authors:** Arpine Ayvazyan, Thomas Stegemann, Mayra Galarza Pérez, Manuel Pramsohler, Serhat Sezai Çiçek

**Affiliations:** 1Department of Pharmaceutical Biology, Kiel University, Gutenbergstraße 76, 24118 Kiel, Germany; 2Botanical Institute and Botanic Gardens, Kiel University, Am Botanischen Garten 1-9, 24118 Kiel, Germany; 3Laimburg Research Centre, 39040 Auer/Ora, Italy

**Keywords:** flavonoids, linolenic acid, linoleic acid, kaempferol, quercetin, ketoglutaric acid, pyruvic acid, glyoxylic acid, camphor, cymene

## Abstract

The herb of *Trigonella caerulea* (Fabaceae), commonly known as blue fenugreek, is used for the production of traditional cheese and bread varieties in the Alpine region. Despite its frequent consumption, only one study so far has focused on the constituent pattern of blue fenugreek, revealing qualitative information on some flavor-determining constituents. However, with regard to the volatile constituents present in the herb, the applied methods were insufficient and did not take relevant terpenoids into account. In the present study, we analyzed the phytochemical composition of *T. caerulea* herb applying a set of analytical methods, such as headspace-GC, GC-MS, LC-MS, and NMR spectroscopy. We thus determined the most dominant primary and specialized metabolites and assessed the fatty acid profile as well as the amounts of taste-relevant α-keto acids. In addition, eleven volatiles were quantified, of which tiglic aldehyde, phenylacetaldehyde, methyl benzoate, *n*-hexanal, and *trans*-menthone were identified as most significantly contributing to the aroma of blue fenugreek. Moreover, pinitol was found accumulated in the herb, whereas preparative works led to the isolation of six flavonol glycosides. Hence, our study shows a detailed analysis of the phytochemical profile of blue fenugreek and provides an explanation for its characteristic aroma and its health-beneficial effects.

## 1. Introduction

*Trigonella caerulea* (L.) Ser. (Fabaceae subfam. Papilionoideae), commonly known as blue fenugreek, is a flowering annual cultivated in the Alps and mountains of eastern and southeastern Europe [1,2]. Unlike the closely related fenugreek (*T. foenum-graecum*), which is a major component in most curry mixtures and therefore well-known and consumed around the world, *T. caerulea* is of regional importance [3,4,5]. Blue fenugreek seeds are used as a spice in Georgian cuisine and the Caucasus region, whereas the young seedlings are eaten with oil and salt [4,5,6,7]. In Switzerland, *T. caerulea* herb is added to the traditional Schabziger cheese in amounts of 2.0 to 2.5%, while it is mixed with flour for the flavoring of bread in South Tyrol, the German speaking part of Northern Italy [4,8]. For bread production, 2 g of blue fenugreek herb are added to 500 g of flour [9]. In South Tyrol, blue fenugreek is stored for a period of up to six months by many traditional farmers before being made commercially available. This measure should increase the aromatic flavor of the herb and the bread made thereof, respectively. The traditional usage of blue fenugreek in the Alps is reflected in its German name, where it is referred to as “Schabzigerklee” or “Brotklee”, but also called by the ethically questionable term “Zigainerkraut”, which means herb of the gypsies.

The few phytochemical studies on *T. caerulea* mostly focused on the seeds, which were analyzed together with the seeds of *T. foenum-graecum* and other *Trigonella* species [3,10,11]. Brenac and Sauvaire [10] investigated sterols and steroidal sapogenins of seven *Trigonella* species and sitosterol was found to be the major sterol in blue fenugreek seeds, with more than 50% of the total sterol content (2 mg/g dry weight). The major steroidal sapogenin was diosgenin, with approximately two-thirds of the total content (8 mg/g dry weight), which is of interest as diosgenin is used for the synthesis of cortisol. Dinu et al. [11] investigated the fatty acid profile and determined α-linolenic acid, linoleic acid, and palmitic acid as the dominant compounds in *T. caerulea* seeds. In the same study, *T. foenum-graecum* seeds were also analyzed, showing a very similar fatty acid pattern. The most recent study on the seeds was conducted by Farag et al. [3], who combined ultra-high performance liquid chromatography coupled to mass spectrometry (UHPLC-MS), and gas chromatography coupled to mass spectrometry (GC-MS) for the analysis of three *Trigonella* species. GC-MS analysis confirmed the aforementioned observed fatty acids, showing several amino acids, keto acids, and other organic acids in blue fenugreek seeds. Liquid chromatography coupled to mass spectrometry (LC-MS) analysis revealed a plethora of flavonoids and saponins, which were, however, not clearly defined. Neither was the origin of these compounds evident, as a seed mixture of three different *Trigonella* species was used.

In contrast to the seeds, the blue fenugreek herb was only investigated once for its phytochemical composition, with the focus on the flavor-determining constituents [8]. Thereby, different chromatographic methods were applied for evaluating the profiles of short-chain fatty acids, aldehydes, and α-keto acids. Of the latter compound class, four constituents were defined as key components, namely pyruvic acid, α-ketoglutaric acid, α-ketoisovaleric acid, and α-ketoisocapronic acid. A recent study quantified the amount of oxalic acid in blue fenugreek herb and other edible plants, such as licorice, sweet potato, okra, cocoa, and different types of legumes [5], and the amount of total oxalate was determined as 1.25% in the dried herb, among which 0.07% were soluble oxalate.

With regard to the characteristic aroma of blue fenugreek, the flavor-determining role of α-keto acids is reasonable, as these compounds are also responsible for the aroma of fermented food products [12]. However, with respect to the volatile constituents, the application of thin layer chromatography, and the limited focus on only aldehydes falls short as also terpenoid constituents can have a great impact on the aroma of plants [8]. Thus, our aim was to investigate the metabolic profile of blue fenugreek herb with a wider scope and up-to-date analytical methods in order to eventually identify additional flavor-contributing constituents. In addition, other specialized (secondary) plant metabolites, e.g., flavonoids, should also be taken into consideration. This is even more important as blue fenugreek is used as a dietary supplement (in combination with kale) and the respective product is standardized on the amount on flavonoid glycosides (>12 mg/g extract) [13].

## 2. Results

### 2.1. Primary Metabolites

#### 2.1.1. Polar Metabolites

Polar primary metabolites were analyzed using a metabolomics approach [14]. The polar fraction of a chloroform-methanol-water extract was investigated by means of GC-MS and untargeted profiling. An example chromatogram and compound list of the commercial samples *T. caerulea* sample 1 to *T. caerulea* sample 3 (TC1–TC3) are provided in the supporting information (Appendix A, Appendix A). Two peaks were found dominating in the chromatogram, namely those of sucrose and the methylated sugar alcohol pinitol (Figure 1). In addition, noteworthy signals were detected for fructose and glucose, the sugar alcohols mannitol, *myo*-inositol, and glycerol, as well as for malic acid, malonic acid, phosphate, and succinic acid.

#### 2.1.2. Fatty Acid Profile

Fatty acid analysis was accomplished by esterification with methanol and subsequent gas chromatography coupled to flame ionization detection and mass spectrometry (GC-FID/MS) analysis. Palmitic acid and α-linolenic acid were found to be the most abundant fatty acids (Figure 2). Moreover, significant amounts of linoleic acid and stearic acid were found in the herb of blue fenugreek, as well as small amounts of arachidic acid, myristic acid, and margaric acid. A sample chromatogram is depicted in Appendix A and results are given in Table 1.

#### 2.1.3. Quantification of α-Keto Acids

Using ultra-high performance chromatography coupled to tandem mass spectrometry (UHPLC-MS/MS) and multiple reaction monitoring, we determined the contents of ten α-keto acids (Figure 3) after conversion into their O-(2,3,4,5,6-pentafluorobenzyl)oxime derivatives by the method described by Noguchi et al. [15]. Among the thus quantified compounds were also the proposed key components pyruvic acid, α-ketoglutaric acid, α-ketoisovaleric acid, and α-ketoisocapronic acid (Appendix A). Additional relevant α-keto acids were retrieved by means of GC-MS and the method of Lee et al. [16]. A total of eleven α-keto acids were found in considerable amounts and commercial standards were therefore obtained. Of the eleven standards, however, oxaloacetic acid did not show useful results and had to be excluded.

The results of the three commercial samples TC1–TC3 are shown in Table 2. In all three samples, glyoxylic acid and α-ketoglutaric acid were the two dominating α-keto acids with amounts of 40 to 86 mg/kg dry plant material. Moreover, high amounts of pyruvic acid (8.1 to 14 mg/kg) were determined, whereas the remaining seven α-keto acids quantified in this study showed concentrations of 0.4 to 4.3 mg/kg. The total content of α-keto acids was found between 115 (TC3) and 185 (TC2) mg/kg, thus showing distinct variations. However, the three samples also differed with regard to their α-keto acid pattern, i.e., α-ketobutyric acid, showing 3 to 5 times higher amounts in sample TC2, or the ratio of the two major components ranging from 1.8 (TC1) to 0.8 (TC3).

### 2.2. Secondary (Specialized) Metabolites

#### 2.2.1. Quantification of Volatile Constituents in Commercial Samples

Volatile constituents of blue fenugreek herb were quantified using headspace-GC-MS/MS and external calibration with eleven compounds, which were found in relevant concentrations (Figure 4 and Appendix A).

The results are given in Table 3 and show three dominating constituents in all three samples, namely tiglic aldehyde, *trans*-menthone, and camphor. The amounts on camphor (10 mg/kg) and *trans*-menthone (8.3 to 8.6 mg/kg) were comparable in all three samples, whereas the concentration on tiglic aldehyde was differing significantly with 8.4, 17, and 6.8 mg/kg, respectively. Other compounds found in considerable amounts were benzaldehyde, phenylacetaldeyhde (hyacinthin), safranal, and bornyl acetate, being present in amounts of 1 mg/kg or above.

Moreover, all quantified compounds (except *p*-cymene in samples TC1 and TC2) showed concentrations above their respective olfactory threshold values (Table 3), and thus contribute to the aroma of blue fenugreek herb. Five compounds were found to predominantly affect the odor of blue fenugreek, namely the aldehydes tiglic aldehyde, phenylacetaldehyde, and *n*-hexanal, as well as methyl benzoate and *trans*-menthone (Table 4). Other compounds with values significantly above their olfactory thresholds were camphor, menthol, and benzaldehyde, as well as *p*-cymene in sample TC3.

#### 2.2.2. Isolation and Identification of Flavonoids

In order to detect eventual non-volatile secondary metabolites, ultra-high performance liquid chromatography coupled to photodiode array detection (UHPLC-PDA) analysis of a crude methanol extract of blue fenugreek herb was performed (Appendix A). Thereby, several flavonoids were detected, showing characteristic UV spectra with absorption maxima at 249 to 265 nm and at 347 to 354 nm, respectively. Additional UHPLC-MS/MS analysis revealed the two major flavonoids to be triglycosides, whereas the minor components showed two sugar moieties. The fragmentation pattern of the flavonoids revealed both hexoside as well as deoxyhexoside moieties, while the remaining aglycone fragments pointed at kaempferol and quercetin scaffolds. As no absolute determination of the structures could be performed without isolating the respective constituents, larger amounts of plant material were extracted and made subjects for a preparative phytochemical work-up.

Repeated chromatographic separation using liquid-liquid fractionation, vacuum liquid chromatography, size exclusion chromatography, and semi-preparative HPLC led to the isolation of the two major flavonoids (**1** and **2**) along with four minor constituents (**3** to **6**). After 1D and 2D nuclear magnetic resonance (NMR) spectroscopic experiments and comparison of the acquired data with the data obtained from the literature, the isolated compounds were identified as quercetin 3-O-(2″-O-α-L-rhamnopyranosyl)-β-D-glucopyranoside 7-O-β-D-rhamnopyranoside (**1**) [25], kaempferol 3-O-(2″-O-α-L-rhamnopyranosyl)-β-D-glucopyranoside 7-O-β-D-rhamnopyranoside (**2**) [25], quercetin 3-O-(2″-O-α-L-rhamnopyranosyl)-β-D glucopyranoside (**3**) [26,27], quercetin 3-O-β-D-glucopyranoside 7-O-β-D-rhamnopyranoside (**4**) [25,28], kaempferol 3-O-(2″-O-α-L-rhamnopyranosyl)-β-D-glucopyranoside (**5**) [26,29], and kaempferol 3-O-β-D glucopyranoside 7-O-β-D-rhamnopyranoside (**6**) [25] (Figure 5, Appendix A).

## 3. Discussion

The first interesting finding of our detailed phytochemical analyses was revealed using metabolic profiling, by which we discovered pinitol as one of two major carbohydrates (Appendix A, Appendix A). The relative concentration was more or less the same in all of the investigated samples and comparable to that of sucrose, the second highly abundant carbohydrate in blue fenugreek herb. A study on the occurrence and accumulation on pinitol in *T. foenum-graecum* found the compound to undergo seasonal variation, thereby showing an increase in content in the leaves during the generative period of plant vegetation [30]. In contrast, the content in the stems remained stable for the whole time of the investigations (56 to 126 days after sowing). Eighty-six days after sowing, which was comparable to the harvesting date of our study, *T. foenum-graecum* showed about half of the amount of sucrose and the same level as glucose in the leaves. Although no quantification of the polar metabolites was conducted in our study, at the same time point, the amount of pinitol in *T. caerulea* was significantly higher than that of glucose, and rather comparable to the level of sucrose. Apart from the role of pinitol in plant carbohydrate metabolism, the compound was also found to exhibit anti-hyperglycemic effects in vivo, which seem to derive from insulin-sensitizing or insulin-mediating properties, respectively [31,32,33].

Other compounds with health-beneficial effects present in blue fenugreek herb were linoleic acid and α-linolenic acid. In all of the investigated samples, the latter compound was found to be one of the two dominant fatty acids, which was only surpassed by palmitic acid (Appendix A, Table 1). The fatty acid composition was also the topic of a comparison study of fenugreek and blue fenugreek seeds, which reported high amounts of linoleic acid and α-linolenic acid in the two species and much lower amounts of palmitic acid [11].

In the same study, the occurrence of flavonoids was also proposed, however, with no details on eventual flavonoid types or sugar moieties. A metabolite profiling study on the seeds of three *Trigonella* species, also showed the occurrence of flavonoids [3], namely C- and O-glycosides of apigenin and luteolin, respectively. However, no assignment of the sugar moieties or eventual linkages were given, indicating the limitations of the metabolomics approach for many secondary metabolites. Furthermore, even more interestingly, no mentions on the occurrence of flavonol derivatives were made, and thus on the flavonoid types detected in our study. Therefore, different scaffolds seem to be present in the seeds and the herb of *T. caerulea*, with the herb containing predominantly di- and triglycosides of kaempferol and quercetin (Figure 5). This is of interest as a commercial dietary supplement containing blue fenugreek and kale leaves (4:1) is standardized on the content of flavonoids [13], even though no reports on the flavonoid composition of *T. carulea* herb have been made so far. This preparation, a hydroethanolic (36%) extract, is intended to prevent skin aging by the antioxidative properties of its ingredients, which was partly demonstrated in a recent study [13]. The identification of the major flavonoids in blue fenugreek by our work might lead to the compounds responsible for the skin-protecting effect and thus lay the basis for future compound-related investigations.

Referring to the study of Ney [8], which so far was the only detailed phytochemical investigation of blue fenugreek herb, the author reported pyruvic acid, α-ketoglutaric acid, α-ketoisovaleric acid, and α-ketoisocaproic acid as being key components of *T. caerulea* herb. Ney [8] determined the α-keto acids after a reduction to the respective amino acids and subsequent ion exchange chromatography. In our work, we chose the conversion of the α-keto acids to their O-(2,3,4,5,6-pentafluorobenzyl)oxime derivatives and quantification via LC-MS/MS (Appendix A, Table 2). We thus quantified most of the α-keto acids described by Ney [8], including the supposed key components. Of those, pyruvic acid and even more α-ketoglutaric acid were indeed found in high amounts in all of the investigated samples. However, α-ketoisovaleric acid and α-ketoisocaproic acid were present in lower amounts and in the range of other keto acids, such as α-ketobutyric acid, α-ketovaleric acid, and α-ketoanteisocaproic acid. In addition, we found high concentrations of glyoxylic acid, being the major α-keto acid in two of three commercial samples (Table 2).

Even more differing than the results of the α-keto acids, were our findings on the aldehyde composition. Using headspace GC-MS/MS, we determined a completely different profile of aldehydes than Ney [8], who was using thin layer chromatography after conversion to the respective dinitrophenylhydrazone derivatives. With tiglic aldehyde, *n*-hexanal, benzaldehyde, phenylacetaldehyde, and safranal, we identified five aldehydes with concentrations above 1 mg/kg in the dried herb (Table 3), which were not mentioned by Ney [8]. Out of these, tiglic aldehyde, phenylacetaldehyde, and *n*-hexanal were found to have a great impact on the aroma of blue fenugreek herb (Table 4). Other compounds clearly affecting the flavor of blue fenugreek were methyl benzoate and *trans*-menthone. With odor activity values above fifty, these five components should also play a role in the aroma of the traditional Schabziger cheese, for which blue fenugreek herb is added in concentrations of 2.0 to 2.5% [8]. For the flavoring of bread in South Tyrol, instead, only tiglic aldehyde and phenylacetaldehyde would reach the respective concentrations [9]. Apart from *trans*-menthone, we also determined high amounts of the monoterpene camphor, with a concentration of around 10 mg/kg in all measured samples (Table 3).

Certainly, the explanation of plant aromas by odor activity values is oversimplified as olfactory thresholds were determined in aqueous solutions and do not necessarily show the same values in other matrices. In addition, synergistic and masking effects can affect flavor perception [17]. Still, the findings of our study contribute to the knowledge on blue fenugreek herb and thereby provide an explanation for its characteristic aroma. As the aroma of the herb is said to be increasing over time, further studies will concentrate on eventual processes occurring during storage.

With regard to the initially mentioned high amounts on oxalic acid (1.25%) and the recommended maximum daily intake of no more than 100 mg oxalates for people affected by hyperoxaluria [5], 320 to 400 g of Schabziger cheese or bread made out of 2 kg blue fenugreek-containing flour would therefore have to be consumed. Although these amounts are rather high, the intake of oxalate for people at risk must not have to be underestimated, especially when additional oxalate-rich foods (spinach, rhubarb), teas (licorice), or dietary supplements are consumed.

To summarize, with the results of our study, new and significant knowledge on the constituent pattern of *T. caerulea*, a plant of growing popularity, is presented. Our findings give explanations for the smell and taste of blue fenugreek and thus for its culinary use. Moreover, with the identification of the major flavonoids, our study reveals the compounds responsible for the supposed beneficial antioxidant effects in nutraceutical preparations.

## 4. Materials and Methods

### 4.1. Plant Material and Chemical Reagents

Dried plant material for isolation and analytical studies (Südtiroler Brotklee Zigainerkraut, Lot 17/2019) was obtained from Feichter Bernhard (Toblach, Italy). Additional samples for analytical studies were purchased from Alfred Galke GmbH (Bad Grund, Germany; Schabziger Bio geschnitten, Lot 34060) and from Lebensbaum (Diepholz, Germany; Schabziger Klee geschnitten und getrocknet). Specimens of all three plant samples (TC1–TC3) are located at the Department of Pharmaceutical Biology in Kiel, Germany.

Pyruvic acid (sodium pyruvate, Lot A0393530) was obtained from ThermoFisher Scientific (Waltham, MA, United States), glyoxylic acid monohydrate (Lot WXBC8625V), 2-ketobutyric acid (Lot BCBW9441), α-ketoisocapronic acid (4-methyl-2-oxovaleric acid, Lot 0000101887), α-ketoglutaric acid (Lot BCBX6537), phenylpyruvic acid (Lot BCBV9597), 4-hydroxyphenylpyruvic acid (Lot BCCC9999), 2-oxovaleric acid (Lot BCCC2216), α-ketoanteisocapronic acid (3-methyl-2-oxopentanoic acid, Lot 19,897-8), α-ketoisovaleric acid (sodium 3-methyl-2-oxobutyrate, Lot 0000000796), oxaloacetic acid (Lot SLBQ4770V), tiglic aldehyde (Lot 06813CE), hexanal (Lot 02429AA), hyacinthin (Lot S21728-164), safranal (Lot 00526EE), cymene (Lot 01805AE), isobornyl acetate (Lot 1091784) menthone (Lot BCCF7127), and O-(2,3,4,5,6-pentafluorobenzyl)hydroxylamine hydrochloride (Lot BCCD2221) were purchased from Sigma-Aldrich (St. Louis, MO, United States). N-methyl-N-(trimethylsilyl)-trifluoroacetaminde (MSTFA, Lot 417263702), benzaldehyde (Lot 50696AJ) and sodium hydroxide (Lot 6771) were obtained from Carl Roth GmbH (Karlsruhe, Germany).

LC-MS grade formic acid, Diaion HP-20, and Sephadex LH-20 were purchased from Sigma Aldrich. Silica gel (40–63 µm) for column chromatography, TLC plates (silica gel 60 F254), acetonitrile and water (both of LC-MS grade), gradient grade methanol, and other (analytical grade) solvents were obtained from VWR International GmbH (Darmstadt, Germany). Water used for isolation was doubly distilled in-house. Dimethyl sulfoxide-*d*_6_ (99.80%, Lot S1051, Batch 0119E) for NMR spectroscopy was purchased from Euriso-top GmbH, Saarbrücken, Germany, and conventional 5 mm NMR sample tubes were obtained from Rototec-Spintec GmbH (Griesheim, Germany).

### 4.2. General Experimental Procedures

Thin layer chromatography for the detection of flavonoids was performed using ethyl acetate–water–acetic acid–formic acid (20:5.4:2.2:2.2) as the eluent and diphenylboryloxy ethylamine–macrogol as the spraying reagent. Pressurized solvent extraction was performed with a Speed Extractor E961 and preparative MPLC was carried out with a Buchi PrepChrom C-700 chromatograph and a Buchi PrepChrom C18 column (250 × 30.0 mm, 15 m particle size) (Büchi, Flawil, Switzerland). Semi-preparative HPLC was accomplished using a Waters Alliance e2695 separations module with an Alliance 2998 photodiode array, and a waters fraction collector (WFC) III (Waters, Milford, MA, USA) using a VP Nucleodur C18 column (250 × 10 mm, 5 µm particle size, Macherey-Nagel GmbH and Co. KG, Düren, Germany).

UHPLC-MS/MS analyses were carried out on a Shimadzu Nexera 2 liquid chromatograph connected to an LC-MS triple quadrupole mass spectrometer using electrospray ionization (Shimadzu, Kyoto, Japan). A Phenomenex Luna Omega C18 column (100 × 2.1 mm, 1.6 µm particle size, Phenomenex, Aschaffenburg, Germany) was employed for the analysis of extracts, fractions, and pure compounds during isolation. Quantification of α-keto acids was accomplished with a Phenomenex Kinetex Biphenyl column (100 × 2.1 mm, 1.7 µm particle size). Headspace GC-MS/MS analyses were performed on a Trace 1310 gas chromatograph equipped with split/splitless (SSL) and programmable temperature vaporizer (PTV) inlets and a TSQ Duo mass spectrometer and the GC-MS instrument used for fatty acid and polar metabolite analysis was a Focus GC gas chromatograph equipped with an SSL inlet, a flame ionization detector and an ISQ mass spectrometer (ThermoFisher Scientific, Waltham, MA, USA). Column used for all GC analyses was a ThermoFisher TG-5SilMS (30 m × 0.25 mm × 0.25 µm). NMR spectra were recorded using a Bruker Avance III 400 NMR spectrometer (Bruker, Rheinstetten, Germany) operating at 400.33 MHz for the proton channel and at 100.66 MHz for the ^13^C channel by means of a 5 mm PABBO broad-band probe with a z gradient unit.

### 4.3. Extraction and Isolation of Secondary Metabolites

A total of 855 g dried and ground plant material was extracted 5 times with a mixture of acetone-water (80:20) using ultra-sonication for 15 min and subsequent maceration for 24 h each. After combining the extracts, the acetone was evaporated and the remaining water was subsequently extracted with ethyl acetate followed by 1-butanol to give 20.5 g and 25.2 g of extract, respectively, after evaporation of the solvents. The remaining water layer yielded 71.2 g. The 1-butanol phase was dissolved in 400 mL water and chromatographed over Diaion HP-20 material, using subsequent elution with water, methanol 25% (400 mL), methanol 50% (600 mL), methanol 75% (600 mL) and methanol 100% (600 mL).

The methanol 25% fraction (1.06 g) was subjected to Sephadex LH-20 using methanol-water (50:50) as eluent. Of the resulting 6 fractions, fraction 2 (0.20 g) was chromatographed with preparative MPLC using water (A) and methanol (B) as solvents with the following gradient: 20% B to 50% B in 60 min, and to 95% B in 30 min. From the obtained 6 fractions (2A–2F), fraction 2B yielded 11.3 mg of compound 1 and fraction 2D yielded 9.5 mg of compound 2, respectively.

The methanol 50% fraction (2.40 g) was also chromatographed with Sephadex LH-20 giving 10 fractions, of which fractions 8 and 9 were further separated by semipreparative HPLC using water (A) and acetonitrile (B) as solvents with the following gradient 15% B to 30% B in 30 min, and to 50% B in 40 min. Thus, 10.1 mg of compound 3 and 10.5 mg of compound 5 were obtained.

The methanol 75% fraction (2.57 g) was as well-subjected to Sephadex LH-20 using methanol-water (50:50) as an eluent and obtaining 12 fractions. Fractions 9 and 10 were chromatographed with semipreparative HPLC in the same manner as before, yielding 15.6 mg of compound 4 and 12.8 mg of compound 6.

### 4.4. Qualitative and Quantitative Analyses

#### 4.4.1. Analysis of Polar Metabolites

Analysis of polar metabolites was accomplished with GC-MS and the method adapted by Fiehn [14]. Briefly, 10 mg of plant material was suspended in 100 µL of water and vortexed for 5 min before adding 300 µL of methanol and 100 µL of chloroform. The extraction was incubated for 1 h over ice and 100 µL of the supernatant (water–methanol) was dried under a stream of nitrogen overnight. An amount of 100 µL of a solution of 20 mg/mL methoxylamine hydrochloride in pyridine was added to the dried extract and incubated at 30 °C for 60 min. Subsequently, 150 µL of MSTFA containing 1% TMCS was added and the reaction was incubated for another 60 min at 45 °C. An amount of 1 µL of the derivatized sample was analyzed by GC-MS with the following parameters: 100 °C hold for 5 min, 25 °C/min to 160 °C hold 1 min, 10 °C/min to 300 °C hold for 12 min. MS parameters: full-scan 50–500 *m*/*z*. Compounds were identified using the NIST database version 2020.

#### 4.4.2. Analysis of Fatty Acid Profile

A total of 50 mg of plant material was analyzed for the fatty acid composition by directly weighing into a 5 mL reaction tube. An amount of 1000 µL of 2.5% H_2_SO_4_ in methanol was added and incubated at 80 °C for 1 h. After cooling at room temperature, 500 µL of *n*-hexane was added, followed by 1500 µL of saturated sodium chloride solution in water. The *n*-hexane phase was transferred into a 2 mL glass vial. An amount of 1 µL was directly injected into the GC-MS. The GC parameters were as follows: 50 °C for 5 min and heated with 5 °C/min to 160 °C hold 1 min, 5 °C/min to 300 °C and hold for 5 min. MS parameters: full-scan from 50–500 *m*/*z*, ion source temperature: 280 °C. Relative quantification was done under the same conditions using FID.

#### 4.4.3. Analysis of Volatile Constituents

An amount of 100 mg of dried and ground plant material was directly weighed into a 20 mL headspace vial, 100 ng of internal standard (toluene-*d*_5_) was added, and the vial was incubated for 30 min at a temperature of 90 °C. An amount of 1000 µL of the headspace was taken with a heated syringe and injected into the GC-MS. The GC program was as follows: 35 °C hold 1 min, 5 °C/min to 120 °C and hold 1 min, 30 °C/min to 300 °C and hold 1 min. The MS parameters were as follows: 43–300 *m*/*z* scan with MS Source at 280 °C. Quantification of compounds was achieved by external calibration using original references standards. Odor activity values for each compound were calculated by dividing the obtained concentration by the respective olfactory activity threshold [34].

#### 4.4.4. Analysis of α-Keto Acids

An amount of 1000 mg of dried and ground plant material was extracted using pressurized solvent extraction (70 °C, 100 bar, 1 min heat up, 5 min hold, 2 min discharge) using one cycle of *n*-hexane and three cycles of methanol, in principle, following the extraction procedure reported by Hidalgo et al. [12]. The methanol extract was evaporated to dryness and reconstituted in 1000 µL of methanol. Following the procedure of Noguchi et al. [15], which was used for quantitation, 5 µL of the solution (or standard solution) was mixed with 45 µL of a mixture of acetonitrile and 0.1% (*w*/*w*) NaOH solution (1:1). An amount of 20 µL of this mixture was treated with a mixture of 10 mg/mL PFBHA and kept at room temperature for 30 min before adding 10 µL of acetone and 100 µL of acetonitrile-NaOH 0.1% (*w*/*w*) solution (1:1). For qualitative analysis, the PFBOximes were further derivatized using MSTFA (after evaporation of the solvent) and analyzed using GC-MS/MS, following the method described by Lee et al. [16], thus retrieving the most relevant α-keto acids for quantitative analysis. For quantitation of α-keto acids, LC-MS/MS was applied in the multiple reaction mode using external calibration with α-keto acid standard solutions over a range of 0.9 to 600 µmol/L. Transitions used for quantification and respective collision energies are given in the supporting information (Appendix A).

## 5. Conclusions

Using a set of analytical techniques, we identified and quantified the key constituents responsible for the smell and taste of *T. caerulea* herb, which is used for flavoring bread and cheese in some regions of the Alps, and which has also become increasingly popular in other parts of Europe. After pyruvic acid and α-ketoglutaric acid were previously determined as important components for the taste of blue fenugreek, we proved glyoxylic acid as another relevant α-keto acid, showing the highest amount in two of the three measured samples. Apart from the importance of α-keto acids, we demonstrated the contribution of several volatiles to the aroma of *T. caerulea* herb, such as tiglic aldehyde, phenylacetaldehyde, methyl benzoate, *n*-hexanal, and *trans*-menthone. Thus, our study also provides an explanation for the characteristic smell of blue fenugreek herb. As a last outcome of our study, the major flavonoids present in *T. caerulea* herb were identified and, subsequently, the value-determining constituents of blue fenugreek nutraceutical preparations.

## Figures and Tables

**Figure 1 plants-12-01154-f001:**
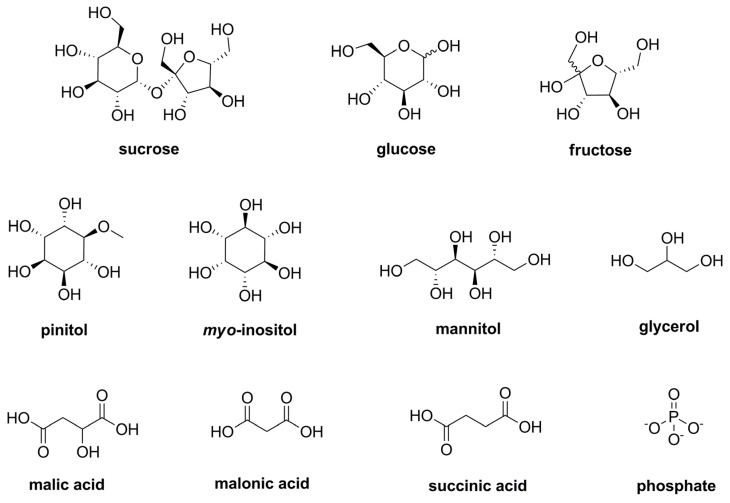
Chemical structures of most dominating primary metabolites.

**Figure 2 plants-12-01154-f002:**
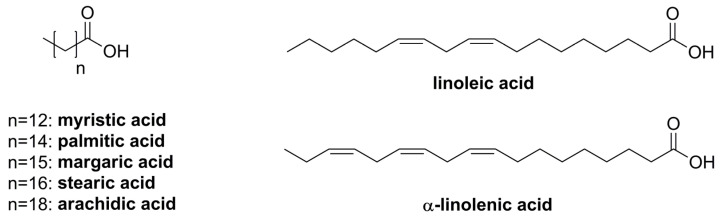
Chemical structures of analyzed fatty acids.

**Figure 3 plants-12-01154-f003:**
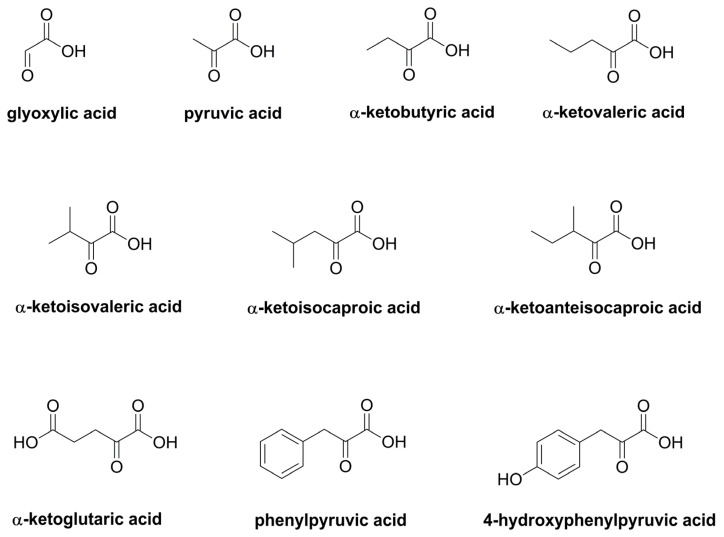
Chemical structures of quantified α-keto acids in blue fenugreek herb.

**Figure 4 plants-12-01154-f004:**
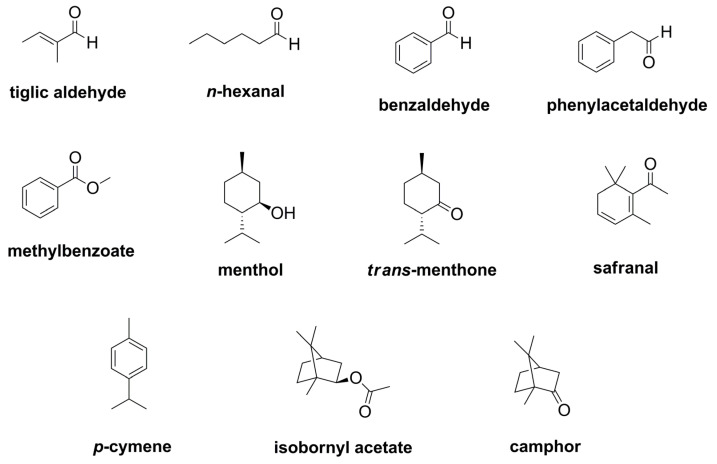
Chemical structures of quantified volatiles in blue fenugreek herb.

**Figure 5 plants-12-01154-f005:**
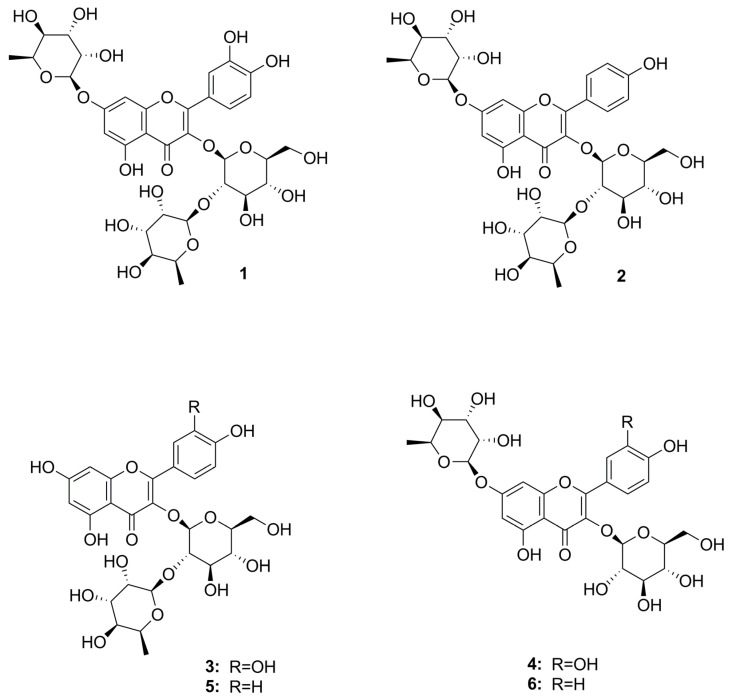
Chemical structures of isolated flavonoids quercetin 3-O-(2″-O-α-L-rhamnopyranosyl)-β-D-glucopyranoside 7-O-β-D-rhamnopyranoside (**1**), kaempferol 3-O-(2″-O-α-L-rhamnopyranosyl)-β-D-glucopyranoside 7-O-β-D-rhamnopyranoside (**2**), quercetin 3-O-(2″-O-α-L-rhamnopyranosyl)-β-D glucopyranoside (**3**), quercetin 3-O-β-D-glucopyranoside 7-O-β-D-rhamnopyranoside (**4**), kaempferol 3-O-(2″-O-α-L-rhamnopyranosyl)-β-D glucopyranoside (**5**), and kaempferol 3-O-β-D glucopyranoside 7-O-β-D-rhamnopyranoside (**6**).

**Table 1 plants-12-01154-t001:** Results of fatty acid analysis. Relative peak areas were calculated from identified compounds and are given in percent.

Compound Name	t_R_ ^1^	RI_exp_ ^2^	RI_lit_ ^3^	TC1	TC2	TC3
myristic acid methyl ester	35.14	1729	1725	1.01	1.51	0.81
palmitic acid methyl ester	39.53	1931	1928	42.35	41.87	43.31
margaric acid methyl ester	41.52	2024	2021	0.78	0.51	0.91
linoleic acid methyl ester	42.90	2094	2093	11.37	12.36	11.26
α-linolenic acid methyl ester	43.05	2112	2108	36.26	35.34	36.11
stearic acid methyl ester	43.43	2134	2127	6.88	7.11	6.47
arachidic acid methyl ester	46.99	2341	2329	1.36	1.31	1.22

^1^ t_R_: Retention time in minutes on the TG-5SilMS GC column. ^2^ RI_exp_: Retention index determined relative to *n*-alkanes (C10–C25). ^3^ RI_lit_: Retention index reported by the NIST database (version 2020).

**Table 2 plants-12-01154-t002:** Contents of the most relevant α-keto acids in commercial samples TC1–TC3. Results are given in mg/kg dried plant material.

Compound Name	TC1	TC2	TC3
glyoxylic acid	74.11 ± 3.14	85.95 ± 2.31	42.59 ± 2.78
pyruvic acid	8.12 ± 0.91	14.48 ± 1.40	9.69 ± 0.75
α-ketobutyric acid	0.83 ± 0.13	4.26 ± 0.02	1.42 ± 0.11
α-ketoisocaproic acid	1.09 ± 0.53	1.29 ± 0.37	1.81 ± 0.31
α-ketoglutaric acid	42.31 ±1.84	68.17 ± 5.26	50.46 ± 0.67
phenylpyruvic acid	0.39 ± 0.14	0.61 ± 0.35	0.41 ± 0.22
4-hydroxyphenylpyruvic acid	0.54 ± 0.33	0.80 ± 0.16	0.68 ± 0.07
α-ketovaleric acid	2.78 ± 0.76	3.28 ± 0.83	1.66 ± 0.97
α-ketoanteisocaproic acid	1.47 ± 0.64	2.56 ± 0.73	1.66 ± 0.71
α-ketoisovaleric acid	3.20 ± 1.07	3.82 ± 0.95	3.53 ± 0.64
total content	134.8	185.2	115.3

**Table 3 plants-12-01154-t003:** Contents of the most relevant volatile constituents in commercial samples TC1–TC3 as well as olfactory thresholds reported in the literature. Quantitative results are given in mg/kg dried plant material and olfactory thresholds are given in ppm in aqueous solutions.

Compound Name	TC1	TC2	TC3	Olfactory Thresholds
tiglic aldehyde	8.44 ± 0.58	16.91 ± 1.75	6.78 ± 0.34	0.01 [17] *
*n*-hexanal	0.36 ± 0.04	0.35 ± 0.02	0.53 ± 0.01	0.0045–0.01 [18,19]
benzaldehyde	1.06 ± 0.05	0.95 ± 0.04	1.04 ± 0.10	0.35 [20]
phenylacetaldehyde	1.02 ± 0.02	1.04 ± 0.01	1.05 ± 0.04	0.004 [21]
methyl benzoate	0.43 ± 0.01	0.44 ± 0.00	0.47 ± 0.02	0.005 [22]
menthol	0.09 ± 0.01	0.09 ± 0.00	0.09 ± 0.02	0.022 [17]
*trans*-menthone	8.41 ± 0.11	8.55 ± 0.09	8.28 ± 0.38	0.17 [18]
safranal	1.25 ± 0.00	1.25 ± 0.00	1.25 ± 0.00	1 [19]
*p*-cymene	0.08 ± 0.01	0.08 ± 0.01	1.53 ± 0.12	0.1 [17]
isobornyl acetate	2.22 ± 0.01	2.22 ± 0.01	2.70 ± 0.02	1.8 [23]
camphor	10.23 ± 0.35	10.30 ± 0.40	10.41± 0.30	1.29 [24]

* Olfactory threshold value was derived from related compounds.

**Table 4 plants-12-01154-t004:** Odor activity values of the most relevant volatile constituents in commercial samples (TC1-TC3) and odor descriptions reported in the literature.

Compound Name	TC1	TC2	TC3	Odor Description
tiglic aldehyde	844	1691	678	green, fruity
*n*-hexanal	50	48	73	green, grassy
benzaldehyde	3	3	3	almond, nutty
phenylacetaldehyde	255	260	263	sweet, rose
methyl benzoate	86	88	94	fruity, sweet
menthol	4	4	4	minty
*trans*-menthone	49	50	49	minty
safranal	1	1	1	herbaceous, sweet
*p*-cymene	<1	<1	15	sweet, citrusy
isobornyl acetate	1	1	2	herb, woody, sweet, minty
camphor	8	8	8	piquant

## Data Availability

Data is contained within the article or Appendix A.

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
