# Peer review of "Phytochemical Profile of Trigonella caerulea (Blue Fenugreek) Herb and Quantification of Aroma-Determining Constituents"

_plants, 2023, doi:10.3390/plants12051154_

Round 1

Reviewer 1 Report

This study investigated the chemical profile of Trigonella caerulea herb applying several analytical methods. This work is of interest, but need of a minor revision to be published.

1) The paragraph that describes plant materials must be improved!!! Authors used different samples for the analysis?? And the botanical identification? A voucher specimen? Please, explain.

2) Discussion section and conclusion must be improved.

Author Response

Dear Reviewer,

This study investigated the chemical profile of Trigonella caerulea herb applying several analytical methods. This work is of interest, but need of a minor revision to be published.

Thank you very much for appreciating our work and for your comments. Please find our answers below:

  • The paragraph that describes plant materials must be improved!!! Authors used different samples for the analysis?? And the botanical identification? A voucher specimen? Please, explain.
  • Yes, we used three commercial samples of Trigonella caerulea. One of the three samples was purchased in higher amounts (1 kg) in order to have enough material for compound isolation. This sample (TC1) was also used for analytical studies. Of the other two samples (TC2 and TC3), we only obtained smaller amounts as we used them only for analytical purposes. As the samples were all commercial dried and cut herb not botanical identification was conducted. However, we kept about 10 g of every sample as specimens at our department. We added this information to the manuscript.
  • Discussion section and conclusion must be improved.
  • We extended the discussion section incorporating recent references, e.g. oxalate content or nutraceutical usage of blue fenugreek. The latter reference was interesting with regard to the flavonoids identified in our study, so we emphasized this point. We also made clearer in the manuscript that blue fenugreek became of increased interest. Likewise, the conclusions were adapted.

We hope that with the changes we’ve made you will find our manuscript now suitable for publication.

Kind regards,

Reviewer 2 Report

Major comment

Article ‘’Phytochemical Profile of Trigonella caerulea (Blue Fenugreek) Herb and Quantification of Aroma-Determining Constituents’’ describe the results of phytochemical composition of Trigonella caerulea investigated using Headspace-GC, GC-MS, LC-MS, and NMR spectroscopy. Authors assessed the fatty acid profile and the amounts of taste-relevant α-keto acids. Authors also identified and quantified tiglic aldehyde, phenylacetaldehyde methyl benzoate, n-hexanal, and trans-menthone as volatile compounds which most significantly contributing to the aroma of blue fenugreek. Article is well done and it is worth to be published in Plants Journal.

 Minor comments

1. Introduction

Line 36: add bracket after Italy

Line 46: moit ‘’of’’ after ‘’content’’

Lines 51, 52, 54: ordinary, abbreviations should be explained when they are mentioned for the first time

2. Results

Lines 74, 83, 95, 145, 159: ’ordinary, abbreviations should be explained when they are mentioned for the first time

Table 2, first column, last row: add total

Line 134: add space after Table 3

4. Materials and Methods

Line 278: explain ‘’WFC’’

Line 288: explain ‘’SSL, PTV,TSQ’’

Line 291: explain ‘’ISQ’’

Line 295: explain ‘’PABBO’’

Line 304: add space before ‘’ml’’ and change ‘’ml’’ into ‘’mL’’ (like in line 302)

Lines 312, 317: add space before ‘’g’’

Lines 324 – 326: change ‘’µl’’ into ‘’µL’’ (like in line 327)

Lines 337, 340: change ‘’ml’’ into ‘’mL’’

 References

References list should be corrected according to journal style (see foods-template). All titles of articles should be write in the same style (for example, compare references no. 10 and 11).

Line 401: put Trigonella caerulea into Italic

Line 427: Manuscript title in all reference should be write on the same way. Please, change.

Line 451: change ‘’Phytochemistry 2000, 53’’ according journal style

 Supplementary material (Supporting Information)

Page 2, Figure S1, note, line 3: change letter ‘’x’’ into into symbol ‘’ב’

Page 3, Figure S2, note, line 3: change letter ‘’x’’ into into symbol ‘’ב’

Page 4, Figure S3, note, line 4: change letter ‘’x’’ into into symbol ‘’ב’

Page 5, Figure S4, note, line 3: change letter ‘’x’’ into into symbol ‘’ב’

Page 6, Figure S5, note, line 3: change letter ‘’x’’ into into symbol ‘’ב’

Table S1: add explanation of ‘’TC’’ below table

Author Response

Dear Reviewer,

Thank you very much for your kind words and for thoroughly reading our manuscript. We addressed all of your comments and conducted the necessary changes, which were as follows:

Minor comments

  1. Introduction

Line 36: add bracket after Italy

  • Bracket was added.

Line 46: moit ‘’of’’ after ‘’content’’

  • „of“ was deleted.

Lines 51, 52, 54: ordinary, abbreviations should be explained when they are mentioned for the first time

- abbreviations of UHPLC-MS, GC-MS, and LC-MS were explained.

  1. Results

Lines 74, 83, 95, 145, 159: ’ordinary, abbreviations should be explained when they are mentioned for the first time

  • Abbreviations for TC1–TC3, GC-FID/MS, UHPLC-MS/MS, UHPLC-PDA, and NMR were explained.

Table 2, first column, last row: add total

  • „total“ was added to table 2.

Line 134: add space after Table 3

  • A space was added after Table 3.

  1. Materials and Methods

Line 278: explain ‘’WFC’’

  • WFC was explained.

Line 288: explain ‘’SSL, PTV,TSQ’’

  • SSL and PTV were explained. TSQ is a brand name and no abbreviation.

Line 291: explain ‘’ISQ’’

  • ISQ is a brand name.

Line 295: explain ‘’PABBO’’

  • PABBO is the name of a specific NMR probe head, which we did not find an explanation for. We have an idea, what it stands for (Probe head with ATMA function and broad band observe), however, as we are not hundred percent sure we prefer not to write it in the manuscript.

Line 304: add space before ‘’ml’’ and change ‘’ml’’ into ‘’mL’’ (like in line 302)

  • Spaces were added and „ml“ was changed to „mL“.

Lines 312, 317: add space before ‘’g’’

  • Spaces were added.

Lines 324 – 326: change ‘’µl’’ into ‘’µL’’ (like in line 327)

  • „µl“ was changed to „µL“.

Lines 337, 340: change ‘’ml’’ into ‘’mL’’

  • „ml“ was changed to „mL“ and „µl“ was changed to „µL“.

 References

References list should be corrected according to journal style (see foods-template). All titles of articles should be write in the same style (for example, compare references no. 10 and 11).

Line 401: put Trigonella caerulea into Italic

  • Trigonella caerulea was put in italic font.

Line 427: Manuscript title in all reference should be write on the same way. Please, change.

  • Manuscript title was changed to tall letters.

Line 451: change ‘’Phytochemistry 2000, 53’’ according journal style

  • Phytochemistry was put to italics and journal volume is now given in bold.
  •  

 Supplementary material (Supporting Information)

Page 2, Figure S1, note, line 3: change letter ‘’x’’ into into symbol ‘’ב’

  • Letter „x“ was changed to symbol „ד

Page 3, Figure S2, note, line 3: change letter ‘’x’’ into into symbol ‘’ב’

  • Letter „x“ was changed to symbol „ד

Page 4, Figure S3, note, line 4: change letter ‘’x’’ into into symbol ‘’ב’

  • Letter „x“ was changed to symbol „ד

Page 5, Figure S4, note, line 3: change letter ‘’x’’ into into symbol ‘’ב’

  • Letter „x“ was changed to symbol „ד

Page 6, Figure S5, note, line 3: change letter ‘’x’’ into into symbol ‘’ב’

  • Letter „x“ was changed to symbol „ד

Table S1: add explanation of ‘’TC’’ below table

  • TC1 to TC3 are explained below the table.

Reviewer 3 Report

comments

Author Response

The presented work deals with In the current work, they used a variety of analytical techniques, including Headspace-GC, GC-MS, LC-MS, and NMR spectroscopy, to examine the phytochemical makeup of the herb Trigonella caerulea. Hence, they examined the fatty acid profile and the quantities of taste-relevant -keto acids, as well as the most prevalent primary and specialized metabolites. Eleven volatile compounds were also measured, and it was determined that tiglic aldehyde, phenylacetaldehyde, methyl benzoate, n-hexanal, and trans-menthone were the ones that significantly influenced the aroma of blue fenugreek. Moreover, the herb was shown to accumulate pinitol, and preparatory work resulted in the separation of six flavonol glycosides.

The manuscript is generally good written; however, I observed some minor grammar and syntax errors, as well as capitalization and punctuation errors throughout the manuscript text

In the following I provide numerous detailed comments, critiques, concerns and suggestions that should be considered before a final decision on the manuscript should be made. Considering my below-given critiques I believe that the revised manuscript will result in a very different version compared with its current state. Therefore, I suggest a re-submission of this work since it generally provides some interesting outcomes.

Sincerely

Dear Reviewer,

Thank you very much for your detailed and critical review. We corrected all language mistakes raised by you as well as additional errors in the manuscript. We also followed your additional suggestions and hope that with the changes we’ve made you will now find our manuscript suitable for publication. Please find our details answers below.

The main criticism points are:

1- There are many grammatical, punctuation, syntax errors, so sever English language editing is needed. For example:

- Line #33 – remove (the) before (Georgian).

à “the” was removed.

- Line #34 – remove (of) after (the herb).

à  The formulation was changed to “T. caerulea herb”

- Line #46 – (two-thirds) instead of (two third )

à changed to “two-thirds”

- Line #49 – (the dominant compounds) instead of (the dominating compounds)

à “dominating” was changed to “dominant”, also in the abstract.

- Line #50 –add (,) after (were analyzed)

à colon was set

- Line #53 – (observing fatty acids and showing)not (observed fatty acids and showed)

à “observed” and “showed” was changed to “observing” and “showing”

- Line #57 –(the seeds, the blue) not ( the seeds, blue)

à “the” was inserted

- Line #59 – (were applied for evaluating) not (were applied evaluating)

“for” was inserted.

- Line #59 –add( , )after (chromatography)

à colon was set.

- Line # 65 – (falls short not ) not (fall too short )……….. Etc.

à “fall too short” was changed to “falls short”

- There are many spelling errors Please review and amend it.

à we tried our best to also detect additional mistakes. Please forgive if there may remain some more.

2- Material and method Lacks recent references

  • We added references for all methods, which were not created by ourselves.

3- What is the experimental design? its not clear

  • After the only phytochemical report on T. caerulea herb was dating back to 1986 and blue fenugreek (and its preparations) show increasing popularity, our approach was to investigate the constituent pattern with up-to-date analytical tools. Even more so, as the mentioned report was not even published in an international peer-reviewed journal and the results are only qualitative. Meaning that the conclusions for e.g. key constituents lack a clear rationale.

4- Most of references are old.

  • We updated the references with more recent works, wherever possible. These comprise e.g. references for the identification of flavonoids as well as for the introduction and discussion.

5- The paper lacks novelty , as there are many papers that have discussed this point in more detail

  • Of course, there are many papers that discuss the phytochemical profile of related plants species, e.g. the well-known Trigonella foenum-graecum. However, as mentioned above, for T. caerulea the only report, in our eyes, was insufficient, especially in hindsight of its increasing popularity.

6- Discussion lacks the supported scientific reasons.

  • We extended the discussion section, also taking more recent references into account. We also tried to discuss the nutraceutical value of blue fenugreek after we found evidence for its use against skin aging.

7- English language needs to be improved.

  • We conducted the requested changes.